# Cellular Senescence in Liver Cancer: How Dying Cells Become “Zombie” Enemies

**DOI:** 10.3390/biomedicines12010026

**Published:** 2023-12-21

**Authors:** Aurora Gazzillo, Camilla Volponi, Cristiana Soldani, Michela Anna Polidoro, Barbara Franceschini, Ana Lleo, Eduardo Bonavita, Matteo Donadon

**Affiliations:** 1Cellular and Molecular Oncoimmunology Laboratory, IRCCS Humanitas Research Hospital, 20089 Rozzano, Italy; aurora.gazzillo@humanitasresearch.it (A.G.); camilla.volponi@humanitasresearch.it (C.V.); eduardo.bonavita@hunimed.eu (E.B.); 2Department of Biomedical Sciences, Humanitas University, 20072 Pieve Emanuele, Italy; ana.lleo@humanitas.it; 3Hepatobiliary Immunopathology Laboratory, IRCCS Humanitas Research Hospital, 20089 Rozzano, Italy; cristiana.soldani@humanitas.it (C.S.); michela_anna.polidoro@humanitasresearch.it (M.A.P.); barbara.franceschini@humanitas.it (B.F.); 4Division of Internal Medicine and Hepatology, Department of Gastroenterology, IRCCS Humanitas Research Hospital, 20089 Rozzano, Italy; 5Department of Health Sciences, Università del Piemonte Orientale, 28100 Novara, Italy; 6Department of General Surgery, University Maggiore Hospital della Carità, 28100 Novara, Italy

**Keywords:** liver cancer, cellular senescence, senescence-associated secretory phenotype, hepatocellular carcinoma (HCC), cholangiocarcinoma (CCA), colorectal liver mestastases (CLM), therapy resistance

## Abstract

Liver cancer represents the fourth leading cause of cancer-associated death worldwide. The heterogeneity of its tumor microenvironment (TME) is a major contributing factor of metastasis, relapse, and drug resistance. Regrettably, late diagnosis makes most liver cancer patients ineligible for surgery, and the frequent failure of non-surgical therapeutic options orientates clinical research to the investigation of new drugs. In this context, cellular senescence has been recently shown to play a pivotal role in the progression of chronic inflammatory liver diseases, ultimately leading to cancer. Moreover, the stem-like state triggered by senescence has been associated with the emergence of drug-resistant, aggressive tumor clones. In recent years, an increasing number of studies have emerged to investigate senescence-associated hepatocarcinogenesis and its derived therapies, leading to promising results. In this review, we intend to provide an overview of the recent evidence that unveils the role of cellular senescence in the most frequent forms of primary and metastatic liver cancer, focusing on the involvement of this mechanism in therapy resistance.

## 1. Introduction

During their lifetime, cells are continuously challenged by various sources of endogenous and exogenous stress [1]. Therefore, they are equipped with sophisticated repair mechanisms that cooperate to reduce the deleterious consequences of damage [2]. One of these defense mechanisms involves the induction of a permanent and irreversible state of cell cycle arrest named cellular senescence. Hayflick and Moorhead, who first observed this phenomenon in 1961, showed that cells undergo a limited number of divisions in response to damage, a mechanism subsequently attributed to telomere shortening [3]. Indeed, senescence prevents the proliferation of damaged cells and the accumulation of further genomic instability, through cell-cycle arrest in the G1 or possibly G2 phase [4,5]. Apart from telomere shortening, other stimuli that trigger senescence are DNA or organelle damage, oncogene activation, loss of tumor suppressor functions, nutrient depletion, and oxidative and genotoxic stress, that may be induced by therapies or even pathogens, including SARS-CoV-2 [6,7].

The variegated types of stimuli that trigger senescence reflect their importance in different cellular conditions [8]. Senescence acts physiologically during embryogenesis, where it occurs at different locations in the mammalian embryo, including the limbs, nervous system, and gut endoderm [9]. Moreover, both pro-apoptotic and pro-growth functions of senescent cells are known to contribute to the preservation or restoration of tissue homeostasis. Cellular senescence can act as a physiological reaction to tissue injury, leading to inflammation, tissue reformation, and remodeling [10]. However, although initial cellular senescence can take part in tissue repair, its prolonged activity can hamper wound healing [11]. Indeed, senescence is typically transient in young healthy tissues, where it contributes to the restoration of tissue homeostasis, while its prolonged accumulation is suspected to be a key driver of pathology [12,13]. Excessive and aberrant accumulation of senescent cells in aging tissues can negatively affect regenerative abilities and generate a proinflammatory environment that can promote the onset of various age-related pathologies, including pulmonary fibrosis, diabetes, and biliary liver damage [14].

Senescence has been observed also in cancer, where it occurs in susceptible neoplastic cells, exerting both antitumor and tumor-promoting features [15]. This mechanism acts as a “double-edged sword” that can function in opposite directions. On one hand, it is a potential mechanism to avoid cell malignant transformation [16]. On the other hand, senescence can also promote cancer development by altering the cellular microenvironment, thus fostering inflammation and tumor escape from therapies [17]. It is known that cellular senescence contributes to the cancer process by taking part in extracellular matrix remodeling, inflammation, invasiveness, angiogenesis, and metastasis development [18]. For this reason, the investigation of cell-intrinsic and cell-extrinsic factors that participate in senescent cells’ switch towards a tumor-promoting role may provide a new horizon for the development of novel therapeutic approaches for cancer treatment, as well as for the understanding of the mechanisms underlying therapy resistance.

In this respect, cellular senescence is now considered an important driving force for the development of chronic liver pathologies, including liver cancer. Indeed, senescence in hepatic cells can arrest cell-cycle progression, giving rise to various phenotype alterations [19]. Senescent cells can undergo a cell death program or start more active proliferation, thus leading to transformation into cancer cells [20]. However, they can also secrete factors which act as inhibitors of inflammation and fibrosis, slowing steatosis and, subsequently, cirrhosis processes [21]. This dual function observed in liver cancer was recently the subject of a study of the development of novel senescence-based therapies. In this review, we provide an overview of the recent findings about senescent liver cells concerning their potential contribution to tumor progression in the most frequent liver cancer forms, with a particular focus on the ability of senescence to promote therapy resistance.

## 2. Cellular Senescence: Origin and Distinctive Features

A stable and irreversible cell cycle is not a characteristic that exclusively belongs to the senescent state. Several other cell conditions, including quiescence, terminal differentiation, and dormancy, display this feature [16]. Therefore, it is difficult to discriminate among these related cellular states also because we lack “gold standard” markers of senescence [22]. One of the main hallmarks of senescent cells is the overexpression of cell cycle inhibitors, such as p16^INK4A^, p21^WAF1^, and their master regulator, p53 [23]. The canonical p53–p21 pathway is frequently triggered by DNA damage, which is the principal cause of senescence stress. When p53 is active, it drives the expression of p21, a cyclin-dependent kinase inhibitor whose function is to inhibit cyclin-cyclin-dependent kinases (CDKs) and to promote the formation of the DREAM complex, resulting in the repression of genes encoding for cycle checkpoints at the G1–S phase transition [24,25]. The DREAM complex is formed by p21 and CDKs, which act by de-phosphorylating different members of retinoblastoma (Rb) tumor suppressor proteins, promoting the generation of the RB/E2F complex and thus inactivating the E2F transcription factor [26]. Upon suppression of CDK2–cyclin E activity, p21 maintains Rb in its hypophosphorylated G1 form, and in complex with E2F, silencing E2F target genes and then blocking the cell into a lasting G1 phase arrest. Moreover, activated p21 can also trigger reactive-oxygen species (ROS) production, one of the main causes of cellular senescence-inducing stress [27]. In addition to the DNA damage response (DDR)-associated senescence, the heterochromatinization of proliferative genes causes a senescent state mainly via the p16–RB pathway [28]. p16 (known as cyclin-dependent kinase inhibitor 2A) prevents the phosphorylation of Rb proteins, forming the RB–E2F complex by inhibiting CDK4/6, which restricts cell proliferation by suppressing cell-cycle gene expression [29,30]. However, p16 might not be a specific or sensitive marker for cellular senescence, since it can also be expressed by non-senescent cells, such as macrophages [31]. Additionally, it has been demonstrated that certain senescent cells do not express p16 [32]. Therefore, upon p21 and p16 chronic activation, the Rb protein is always maintained in a hypophosphorylated state by CDKs’ inhibition, leading to E2F transcription factors’ neutralization, thus locking the cell into an indefinite proliferative state. Interestingly, it has been shown that p21 is mainly activated early during the evolution of senescence, whereas p16 seems to maintain it [15].

The senescent cell state is mainly triggered by the chronic activation of the DNA damage response, which is also associated with structural and/or functional defects in cellular organelles [8]. The plasma membrane composition may change in a senescent state. Indeed, upregulation of caveolin-1, an important component of cholesterol-enriched microdomains referred to as caveolae, is one of the most striking structural changes in senescent cells [11]. Moreover, there is evidence of changes in the expression of other plasma membrane proteins during the switch to senescent state, such as receptor-type tyrosine protein phosphatase DEP-1, β2-macroglobulin, and vimentin [33]. Generally, senescent cells display different morphologies compared to healthy cells. Indeed, they are also characterized by large nuclei, multi-nuclei, and chromatin reorganization [8]. Dikovskaya et al. showed that senescent human fibroblasts display multinucleation or enlarged nuclear size caused by incomplete mitosis and failed cytokinesis in vitro [34]. These structural nuclear alterations are mainly associated with the loss of lamin-B1, a protein of the nuclear lamina [35]. Besides, senescent cells may show an enlarged cell body because of their inability to perform cell division, associated with the enhanced formation of actin stress fibers, which can be due to the upregulation of cofilin 1 protein [36]. Moreover, the cytoplasm of senescent cells is characterized by vacuolization, granularity, and intracellular debris, as well as by abnormal morphological and functional changes in organelles. Mitochondria display a swollen morphology, together with a decreased mitochondrial membrane potential, typically accompanied by the increased production of oxygen free radicals [37,38]. Another organelle aberration displayed by senescent state is the expansion of the endoplasmic reticulum, regulated by the transcription factor, ATF6α [39]. Furthermore, the accumulation of lipid-containing granules, known as lipofuscin, reflects senescent lysosomal and autophagic abnormalities [40]. Finally, senescent cells are characterized by the increased secretion of lysosomal enzyme, “senescence-associated β-galactosidase (SA-β-Gal)”, which is often exploited to label senescent cells, both in vivo and in vitro [41].

One of the most distinctive features of senescence is the secretion of several families of soluble and insoluble factors, that can affect surrounding cells by activating various cell-surface receptors and corresponding signal transduction pathways (Figure 1) [42]. These factors can be enclosed within microparticles to allow their delivery to the extracellular environment. This senescence-associated secretory phenotype (SASP) can be triggered by the activation of numerous cellular pathways, including cyclic GMP–AMP synthase, STING, NF-κB, JAK-STAT, NOTCH, and mTOR signaling [16]. This SASP involves the secretion of soluble signaling factors, which include pro-inflammatory cytokines, chemokines, and growth factors, as well as secreted insoluble proteins/extracellular matrix (ECM) components, such as matrix metalloproteinases (MMPs) [43]. The SASP is not static but can also change over time, depending on the intracellular and extracellular environments that shape its abundance and release. Moreover, the SASP may exert opposite effects in the surrounding environment, providing a potent mechanism by which senescent cells can alter the nearby tissues. Indeed, SASP proteases can turn membrane-associated receptors into soluble proteins, degrade and cleave signaling molecules, and/or alter the extracellular matrix [44]. In some cases, the SASP releases pro-inflammatory, proapoptotic, and pro-fibrotic factors, which can skew previously non-senescent cells to become senescent [45,46]. This proapoptotic condition leads to senescent cells’ accumulation and can have damaging effects both locally and systemically [47]. Alternatively, the SASP can release growth and other regenerative factors, limiting apoptosis and fibrosis, thus having positive effects on the neighboring cells and tissues [48]. Finally, analysis via bulk RNA sequencing (RNA-seq) of human and mice fibroblasts revealed differences in transcriptomic signature and SASP based on the cell type, the stage of senescence, and the type of inducers, suggesting that these conditions may lead to differences in function [49]. Interestingly, individual senescent cells inside the same human fibroblast culture showed cell-to-cell transcriptional differences [50]. Therefore, more research exploiting high-resolution methods, including single-cell RNA sequencing, is required to elucidate the characteristics of senescent cells under different conditions in vivo.

Senescent cells can resist cell death via the upregulation of pro-survival and antiapoptotic pathways. Apoptotic resistance is due to persistent activation of BCL-2 antiapoptotic proteins, together with the inhibition of BAX pro-apoptotic proteins by epigenetic repression [51,52]. Moreover, senescent cells have been shown to upregulate several pro-survival pathways, including PI3K–AKT signaling, SRC kinases, heat shock protein (HSP), serpines, and mitochondrial pathways [53]. Hence, senescent cells do not replicate as they did before, but at the same time they refuse to die, despite the cytotoxic environment that they generate. This dual ability of senescent cells to simultaneously block replication and escape death has inspired recent papers to start referring to them as “zombie cells” [54]. This new representation is also supported by their controversial behavior in aging, tissue repair, and, especially diseases, such as cancer. Indeed, senescent cells can have beneficial or detrimental roles that depend on the varying physiological and pathological contexts in which they act [8].

## 3. A Controversial Behavior in Cancer: When Senescent Cells Become “Zombies”

Although cancer cells are the most proliferative cells in mammalian tissues, senescent cells are frequently observed in tumors [55]. Cancer cells generate a harsh environment, which is characterized by hypoxia and nutrient deficiency, a condition easily associated with cellular damage [17]. Moreover, the high replication burden of cancer cells can generate genomic instability, which may be the main driver of senescence [56]. Hence, among the main insults that induce oncogenic senescence, there is DDR signaling, which leads to oncogene activation and downregulation of tumor suppressors [16]. The activation of oncogenes, such as HRAS^V12^ or BRAF, triggers growth arrest, referred to as oncogenic-induced senescence (OIS), which was first described in 1997 [56,57]. Likewise, loss of a tumor suppressor gene, such as Pten, can induce cellular senescence, referred to as PTEN-loss-induced cellular senescence (PICS) [58]. Moreover, senescence can be triggered by other oncogenic pathways, including hyperactivated MYC and WNT–β-catenin signaling [59]. Furthermore, exposure to carcinogens triggers the production of ROS, which represents one of the main causes of DNA damage, initiating p53–p21 signaling [60]. Finally, ROS can induce senescence independently of the DDR by activating the p16–RB pathway [8].

Conventional anticancer therapeutics, such as chemotherapy or radiotherapy, have been shown to induce senescence in cancer cells: this process is known as therapy-induced senescence (TIS) [61]. On one hand, senescence can contribute to antitumor effects and treatment outcomes; on the other, chronic accumulation of senescent cancer cells can stimulate relapse and metastasis [62]. Exposure to chemotherapy drugs or radiation has been shown to increase the presence of senescent cells in malignant and non-malignant tissues via the activation of p53-RB pathways [63,64]. The correlation between therapies and senescence is likely due to the evidence that anticancer treatments can cause DNA damage, although this phenomenon can particularly depend on drug dosage [65]. Indeed, low doses of chemotherapy drugs have been observed to trigger a senescent cell state, while higher doses usually induce apoptosis. Moreover, the thresholds to enter senescence seem to strictly depend on the cell type and to vary between malignant and non-malignant cells [66,67]. This evidence may explain why often only a small subset of tumor cells undergoes senescence upon chemotherapy or radiotherapy treatments. Additionally, chemotherapy is administered systemically; therefore, it may potentially induce senescence at multiple locations, even in non-malignant tissues. Radiotherapy is exploited for the treatment of multiple cancers as it can generate DNA damage via the p53–p21 pathway, inducing apoptosis and senescence [68]. Contrarily to chemotherapy, it has the advantage of local treatment delivery, and thus it is less deleterious for normal tissues surrounding the tumor, although it does not act in a cancer-specific manner [69]. Finally, the presence of senescent cells has also been observed in cancer patients treated with targeted therapies. CDK4/6 inhibitors, for example, act by mimicking the mechanism of action of p16, and thus cellular senescence has been considered a possible outcome of the treatment [70]. Moreover, antiangiogenic drugs have been shown to induce senescence in preclinical models of renal and colorectal cancer, as well as to induce the SASP phenotype in cancer patients [71]. Interestingly, even treatment with monoclonal antibodies has shown to induce senescence. Recent evidence demonstrated that the CD20-targeting monoclonal antibody, rituximab, can promote senescence in lymphoma cells in vitro [72]. Taken together, these data show that senescence is a frequent mechanism induced by multiple anticancer drugs, although different aspects of this phenotype still need to be elucidated in cancer. Additional studies are required to clarify the senescence induction of non-malignant cells as a side effect of anticancer therapies and to understand if senescent cells exert a beneficial or detrimental role in tumor clearance and organ regeneration in patients with cancer.

Senescent cells can perform both a positive and negative role in tumors. They can act as allies against cancer by preventing tumor initiation through the suppression of cell-cycle in damaged cells [8]. This intrinsic mechanism allows the avoidance of the passing of carcinogenic mutations to the next generation of cells, as well as the improvement of the immune clearance of potential tumor cells. In OIS, mutated oncogenes drive various aberrant signaling pathways and DNA replication that, if not immediately arrested, may lead to tumor initiation and development [73]. In response, cells activate senescence, which allows them to induce a firm proliferation arrest, highlighting the crucial protective role of this mechanism against tumor growth [74]. Indeed, senescence may suppress cancer through the extrinsic release of SASP factors, which may reinforce the senescent growth arrest and/or promote immune surveillance. Senescent cells secrete the inflammatory cytokine, IL-1α, which leads to the expression of IL-6 and IL-8 [75]. These inflammatory cytokines increase ROS production and reinforce DDR, thus enhancing cellular senescence in an autocrine fashion [76]. Additionally, IL-6 and IL-8 can spread senescence in the surrounding cancer cells in a paracrine way, helping further tumor suppression [45]. Furthermore, mounting evidence supports the crucial role of senescence in promoting cancer immunosurveillance. SASP factors IL-6, IL-8, and CCL2 induce the recruitment of various antitumor immune cells, such as natural killer (NK) cells, M1 macrophages, and T helper 1 cells, which can exert a tumor clearance function [77]. Finally, immune cells can trigger cancer cell senescence via the secretion of inflammatory cytokines. For example, a study performed on lymphoma cells highlighted that tumor-associated macrophages were able to secrete inflammatory cytokines, promoting TGFβ, that was able to induce senescence in malignant cells, enhancing tumor suppression [78].

Despite all the beneficial roles exerted by senescence in tumors, its persistency at the tumor site can be extremely detrimental: this is the moment when a senescent cell becomes a dangerous “zombie” [46]. Excessive accumulation of senescent cells in tissues can inhibit the regenerative capacities and create a proinflammatory milieu favorable for tumor establishment (Figure 2) [79]. Indeed, a huge variety of SASP factors is associated with pro-tumorigenic mechanisms [80]. The chemokine, CXCL1, has been shown to induce tumorigenesis, while the cytokine, CCL5, has been reported to enhance tumor proliferation [81,82]. Senescent cells can have a role in metastatic growth and invasion, especially via tissue remodeling activity exerted by SASP factors [16]. For example, the ECM degradation induced by MMPs has been shown to promote growth factor release, which can lead to tumor development and evasion [83]. Moreover, the SASP cytokines, IL-6 and IL-8, have been shown to drive the transcription of both MMP genes and epithelial-to-mesenchymal transition (EMT), thus enhancing tumor invasion [84]. Soluble E-cadherin secreted by senescent melanoma cells has been shown to enforce their invasive activity, both in vivo and in vitro [85]. SASP factors are also involved in pro-tumor angiogenesis. MMPs induce tumor neo-vascularization by leading to the release of vascular endothelial growth factor (VEGF) in the TME [86]. Similarly, CXCL5 has been demonstrated to increase blood vessel density associated with metastasis development in CRC [87]. In addition to angiogenesis, another pro-tumorigenic effect of SASP factors is the recruitment of immune suppressive cells to the tumor microenvironment. IL-6-secreting senescent cells were shown to recruit myeloid-derived suppressor (MDSCs) cells in the TME, which then can block immune surveillance by inhibiting CD8+ T cells [88]. SASP factors, Il-1a and IL-8, allow MDSCs to inhibit NK cells and M1 macrophage recruitment, taking part in tumor immune escape [16]. Surprisingly, some cancer therapies that induce senescence have highlighted a possible role of senescent cells in driving tumor cell stemness. Indeed, Milanovic et al. reported that senescent cells can acquire features of stemness, partly through the activation of Wnt signaling [89]. This condition may lead to cancer stem cells, which may likely induce cancer initiation and metastasis [90]. Similarly, senescent fibroblasts in colorectal cancer have been shown to release SASP factors, HGF, MIF, and CCL2, which through the activation of Wnt signaling can increase the tumor cell stemness [91].

The stem-like state triggered by senescence may also be responsible for the emergence of drug-resistant, aggressive tumor clones [92]. Indeed, senescent cancer cells are highly resistant to chemotherapy due to their non-proliferative nature of being in a state of permanent cell cycle arrest [93]. Moreover, therapy-induced senescent cells have been shown to trigger a paracrine secretion of factors able to protect surrounding cells from being killed by the same therapeutic agents [94]. Additionally, senescent cells play an important role in extracellular matrix remodeling, where they promote therapy resistance via the upregulation of ligands and ECM components; this process is known as cell adhesion-mediated drug resistance [95]. In triple negative breast cancer, tumor cell senescence has been correlated with resistance to chemotherapy [96]. Furthermore, it has been shown that the efficacy of immune checkpoint inhibitor (ICI) strategies is impacted by therapy-induced senescence. Accordingly, increased inflammatory cytokines, such as SASP factors, IL-6 and IL-1β, have been associated with poor responses to ICI treatments [97]. Senescence can alter immune cell function and abundance, other than the levels of inflammatory cytokines, resulting in a dysfunctional immune response and an unbalanced inflammatory status, thus leading to an impairment in ICI response. Interestingly, besides cancer cell senescence, therapy resistance has also been associated with a newly discovered attractive mechanism, known as immunosenescence [98]. Indeed, the inflammatory environment generated by tumor development may induce innate and adaptive immune dysfunction, limiting therapy, especially the ICB response. Thus, the existing immunological techniques and experimental progress must cooperate with research elucidating the senescence mechanism, to develop therapies able to fully address the complexities of the TME.

Despite all the results already achieved, the mechanisms underlying the ability of senescent cells to be pro- or anti- tumoral still need to be elucidated, together with the identification of the stage at which senescent cells might become malignant. Therefore, the clarification of the aspects that associate senescence with tumorigenesis will probably allow the development of more efficient therapies to improve patients’ management and treatment.

## 4. The Role of Cellular Senescence in Liver Cancer

The liver is a central organ performing vital functions related to the metabolism of nutrients, toxins, and drugs, as well as digestion and immunity. Because of its anatomy, the portal circulation constantly exposes the liver to a significant number of foreign molecules including pathogens [99]. The portal vein drains blood from the intestine, spleen, and pancreas to the liver, carrying many foreign molecules derived from both food and the intestinal microbiota. Consequently, the high workload and infectious risks make this organ prone to injuries, which can trigger cell death programs, including senescence. Therefore, the hepatic immune compartment is characterized by a delicate balance between tolerance toward harmless antigens and activation against potential pathogens [100]. Despite the liver’s ability of self-regeneration, inflammation and fibrosis can lead to its failure or malignant progression (Figure 3) [101]. Interestingly, senescence does not act only as an age-related biological process, but can have a pivotal role in chronic liver diseases. In liver cancer, cellular senescence acts as a “double agent” whose mechanisms in tumors still need to be completely understood, to be further exploited for the development of more efficient anticancer therapies.

### 4.1. Senescence in Hepatocellular Carcinoma (HCC)

Hepatocellular carcinoma (HCC) is the most prevalent form of liver cancer and ranks as the fourth leading cause of cancer-related mortality globally [102]. Regrettably, HCC is often diagnosed at advanced stages, leaving approximately 70% of patients ineligible for surgical or transplant-based treatments [103]. Indeed, the therapeutic strategies can be limited by patients’ basal clinical conditions, such as cirrhosis [104]. The available therapy options for HCC offer limited benefits, with a 5-year survival rate of less than 20% [105]. Besides surgery, which represents the ideal method for HCC eradication, loco-regional treatments, including radiofrequency and trans-arterial chemoembolization (TACE), allow good results to be obtained in terms of efficacy with limited damage for the most fragile patients [104]. However, the frequent failure of these therapeutic options orientates clinical research to the investigation of new drugs. In recent years, cancer immunotherapy has shown promising results in providing new avenues for HCC treatment. However, immune checkpoint blockade (ICB) therapy, while a breakthrough in HCC, still achieves scarce response rates [106]. Additionally, ICB treatments have been associated with immune-related hepatotoxicity and poorer outcomes in some HCC patients [107]. Indeed, the liver possesses a complex immunological network that creates a delicate balance between tolerance toward harmless antigens and activation against potential pathogens [99]. HCC usually develops in the context of chronic inflammation and cirrhosis, a primary cause of immune exhaustion that, in turn, enhances the liver’s immunosuppressive status. Liver tolerance is indeed mediated by several non-parenchymal cells, which include specialized antigen-presenting cells (APCs), such as dendritic cells (DCs), Kupffer cells (KCs), and other non-immune cells, such as liver sinusoidal endothelial cells (LSECs) and hepatic stellate cells [108]. These cells can induce a tolerogenic environment via the secretion of immunosuppressive functions, such as transforming growth factor (TGF)-β and interleukin (IL)-10, or via the expression of surface inhibitory ligands, including programmed death (PD)-ligand-1 (PD-L1) [109]. Besides immunotherapy, HCC is also associated with chemotherapy, radiotherapy, and targeted therapy based on the use of inhibitors of tyro-sine kinase (TKI) resistance [110]. The poor outcome of HCC patients is indeed mainly due to the high refractoriness of this tumor. However, the mechanisms underlying resistance still remain poorly understood.

Seeking alternative or complementary strategies to tackle HCC, senescence has started to be investigated. This mechanism has been explored as an inducible defense mechanism, which can trigger a stable proliferation arrest in cancer cells and/or immunosurveillance to eliminate senescent or non-senescent pre-malignant cells to obstruct HCC development. Mudbhary et al. showed that p53 mutation causes the inhibition of senescence, which was associated with tumor progression in both zebrafish and human HCC [111]. Indeed, restoring p53 activity reactivates the senescence program, triggering the clearance of HCC cells by the immune system [112]. In a mouse model of fibrotic HCC, senescent hepatic stellate cells (HSCs) showed an anti-fibrotic effect by reducing the extracellular matrix composition, stimulating matrix-degrading enzyme expression, and recruiting NK cells, that cleared activated HSC responsible for fibrosis [113]. Moreover, the induction of a senescent state in HSCs via adenoviral infection alleviated fibrosis in an alcohol-induced fibrosis mouse model [114]. Furthermore, oncogene-induced senescence has been shown to suppress HCC tumor progression [115]. Indeed, senescent pre-malignant hepatocytes have been shown to limit cancer development via the SASP-induced recruitment of immune cells, which is able to eliminate senescent hepatocytes and prevent malignant transformation. In a HCC mouse model, oncogene-induced senescence exerted a protective role against tumor growth [116]. Accordingly, recent evidence on a mouse model of chronic hepatitis B virus (HBV)-induced HCC uncovered a new viral mechanism that enhances hepatocarcinogenesis by deregulating senescence mechanisms [117]. Finally, p53 reactivation in p53-deficient tumors has been shown to lead to complete HCC regression by inducing a cellular senescence program [112]. Interestingly, re-activation of senescent mechanisms induced differentiation and upregulation of inflammatory cytokines, triggering an innate immune response that targeted the HCC cells in vivo, thereby contributing to tumor clearance.

Besides its antitumor effects, senescence may also be an enhancer of HCC carcinogenesis. In a rat HCC model induced by the carcinogen, diethyl nitrosamine (DEN), injection, the appearance of cellular senescence progressively increased during hepatocarcinogenesis. Indeed, the authors observed an increased expression of p16, CDK4, and β-galactosidase, inversely related to cyclin D1, p53, and p21 involvement, indicating that senescence is activated via the p16 pathway during hepatocarcinogenesis induced by DEN [118]. In NASH-related HCC, the accumulation of senescent cells was shown to favor the occurrence of the disease [119]. Moreover, higher expression levels of four cellular senescence-related genes, *EZH2*, *G6PD*, *CBX8*, and *NDRG*, have been associated with the increased migration and invasion of HCC, as well as with a worse prognosis in patients [120]. Accordingly, multiple studies have demonstrated an association between senescence and HCC invasiveness and migration. Galectin-3, a senescence-related multifunctional protein belonging to the β-galactoside-binding protein family, showed a close correlation with vascular invasion and poor survival in a large-scale study of HCC patients [121]. Mechanistically, Galectin-3 activates the β-catenin/TCF4 transcriptional complex via the PI3K-Akt-GSK-3β-β-catenin signaling cascade. This transcriptional complex directly targets IGFBP3 and vimentin, leading to the promotion of angiogenesis and EMT, therefore enhancing HCC invasiveness. Moreover, Lv et al. proved that the deubiquitinase, (DUB) 26S proteasome non-ATPase regulatory subunit 14 (PSMD14), an enzyme involved in the senescence mechanism, is significantly upregulated in the tissues of HCC patients, and acts by enhancing vascular infiltration, and tumor metastasis and recurrence [122]. In addition, PSMD14 can promote phenotypic changes in HCC cells in vitro, fostering cell proliferation invasion and migration. Furthermore, it has also been demonstrated that senescence may promote stemness, tumorigenesis, and therapy-resistance in the HuH-7 HCC cell line [123]. Interestingly, the role of therapy resistance in HCC exerted by senescence can be associated with the reprogramming of HCC immune cell composition. In HBV-associated HCC patients, it has been recently demonstrated that the senescence mechanism reduces antitumor NKT cells’ infiltration, thereby affecting patients’ prognosis [124]. Huang et al. showed that hepatic SASP can promote hepatocarcinogenesis through the bcl3-dependent activation of pro-tumor macrophages, thus promoting HCC progression [125]. Similarly, Wu et al. revealed that the overexpression of senescent gene, EZH2, was positively correlated with the critical gene markers of TAMs, M2 macrophages, M1 macrophages, and monocytes, which exerted a pro-HCC role [126]. G6PD, another senescence-associated gene, has been associated with the development of a HCC-immune-suppressive microenvironment. Indeed, G6PD is not only strongly correlated with immune markers of M2 macrophages but also promoted pro-tumor M2 macrophage polarization through the WNT signaling pathway [127]. Starting from the aforementioned evidence, in recent years, many studies have constructed HCC prognostic models based on genes associated with different forms of cellular senescence. Li et al. established a prognostic signature of senescence-associated genes to predict the prognosis and therapeutic response of HCC patients [120]. The authors demonstrated that the mRNA expression of EZH2, G6PD, LGALS3, and PSMD14 senescence-related genes was elevated in cancer tissues from HCC patients compared with normal gastric cancer parietal tissues, suggesting that these genes may act as potential biomolecular markers for the prognosis and stratification of HCC patients. Similarly, immune senescence profiles have been shown to possibly predict HCC prognosis in liver transplant patients [128]. Overall, the realization of a prognostic model for cellular senescence risk score can not only be exploited to predict HCC patients’ prognosis, but can also provide more strategies for the clinical treatment of HCC patients. Accordingly, recent studies reveal that senescence-related gene signatures can also be used to predict the immunotherapy response [129]. However, further experiments and clinical data are needed to validate the involvement of the senescence mechanism in HCC progression and prognosis.

The administration of conventional therapies for HCC treatment may induce cellular senescence. For instance, Qu et al. showed that the chemotherapeutic agent, cisplatin, triggers senescence mechanisms in HCC cell lines. Cellular senescence induced by cisplatin is dependent on p53 and p21 but not p16 activation [130]. Cisplatin-induced accelerated senescence depends on intracellular ROS generation, which leads to slower tumor growth. This functional link between intracellular ROS generation and cisplatin-induced accelerated senescence may be used as a potential target of HCC. Moreover, cellular senescence is considered as a complication of radiotherapy treatment following the activation of the DDR [131]. Indeed, ionizing radiation induced the long-term expression of senescence markers, p21 and p16^INK4a^, in mice, independently from p53 mutation and immune status [132]. Similarly, a single dose of 25 Gy radiation induced hepatocyte senescence in rats. Several markers of cell senescence were upregulated in hepatocytes after radiation, including SA-β-gal, with an increase in cell size, upregulation of p16 and p21, and the activation of SASPs, such as IL6 and IL1α [133]. Although TIS is a mechanism to control cancer growth, blocking tumor cells’ proliferation, sometimes it can lead to prolonged survival of a subgroup of cancer cells, which can re-enter the cell cycle, increasing stemness gene expression and thus enforcing therapy resistance. Indeed, residual cells after TIS with an increased cancer stem cell phenotype may have profound implications for tumor aggressiveness and disease recurrence. Karabicici et al. showed that senescence induced by the chemotherapy agent, doxorubicin, increased the tumorigenicity and stemness in a HCC cell line, leading to a significant increase in the expression of reprogramming genes, SOX2, KLF4, and c-MYC as well as liver stemness-related genes, EpCAM, CK19, and ANXA3, and the multidrug resistance-related gene, ABCG2 [123]. Therefore, further studies are essential to carefully dissect the balance between the beneficial and detrimental aspects of TIS in HCC, in order to exploit it for the development of effective treatments.

### 4.2. Senescence in Cholangiocarcinoma (CCA)

Cholangiocarcinoma (CCA) is the second most common type of primary liver cancer after HCC, accounting for 10-15% of all primary liver malignancies [134]. CCA includes a cluster of highly lethal and heterogeneous malignant tumors, which can arise at various points of the biliary tree [135]. Indeed, CCAs are divided into three subtypes depending on their anatomical site of origin: intrahepatic (iCCA), perihilar (pCCA), and distal (dCCA) CCA [136]. The high mortality of CCA is due to its silent presentation, together with its aggressive nature and resistance to chemotherapy [137]. Moreover, patients with CCA are frequently asymptomatic in the early stages of the tumor, and around 70% of them are diagnosed at advanced phases when the malignancy is widespread [138]. Inefficient diagnostic methods, together with the extremely heterogeneous genomic, epigenetic, and molecular features of this tumor severely compromise patients’ management and the efficacy of the available therapies [139]. Currently, only 30% of patients are treated with surgical resection or liver transplantation, which represent the only potentially curative options [140]. For unresectable cases, palliative treatments include chemotherapy and/or immunotherapy, which provide a median overall survival of 12 months [138]. CCA is a highly infiltrated tumor, in which cancer cells undergo a complex crosstalk with the stromal compartment [141]. The CCA stroma comprises extracellular matrix proteins, endothelial and mesenchymal cells, as well as cancer-associated fibroblasts (CAFs) [142]. Once recruited and activated by CCA cells, the stromal compartment in turn releases a wide variety of paracrine signals, including chemokines, growth factors, and proteases, to shape the tumor microenvironment [143]. Especially, CAFs have been shown to foster CCA proliferation and drug resistance by releasing signaling molecules and ECM proteins, as well as by recruiting myeloid-derived suppressor cells via IL-6 signaling [144,145]. Moreover, intrahepatic iCCA displays a very limited infiltration of cancer-targeting immune cells and reduced immune response [146]. Accordingly, scRNA-seq analysis showed the abundant infiltration of hyperactivated CD4+ Tregs in iCCA tumors along with reduced CD8+ T-cell effector functions [147]. Therefore, although CCA is mainly considered a low-infiltrated tumor, immune players exert a key role in its pathogenesis [141]. However, most current 2D and 3D in vitro models of CCA fail to recapitulate its TME. Interestingly, a recent research effort developed a 3D organ-on-chip CCA platform, able to integrate the major non-immune components of the TME and the T cell infiltrate, thus reflecting the in vivo CCA niche and tumor drug response [148]. Overall, the crosstalk between tumor and its stroma is considered a key mechanism for cancer progression and metastasis, although the exact mechanism by which CCA acts at the cellular level still needs to be elucidated to improve CCA therapies and overcome drug resistance.

Recent evidence revealed that cellular senescence may be involved in the pathophysiology of cholangiocarcinoma [149]. Sasaki et al. reported that the aberrant expression of polycomb group protein, EZH2, was associated with the development of cholangiocarcinoma by inhibiting senescence in large bile ducts [150]. Specifically, the expression of senescence-related protein, p16^NK4a^, was high in the first phases of cholangiocarcinogenesis, while it decreased in the late stages of the invasive carcinoma. Contrarily, EZH2 expression showed a stepwise increase during tumor growth. These results were explained showing that EZH2 was able to induce the hypermethylation of the p16^INK4a^ promoter, thus inhibiting cellular senescence. This evidence was confirmed via immunohistochemistry of livers with iCCA, as well as by using in vitro assays performed exploiting two CCA cell lines (HuCTT-1 and TFK-1) [149]. In cultured CCA lines, knockdown of EZH2 led to decreased p16^INK4a^ methylation and decreased binding of EZH2 to the p16^INK4a^ gene promoter, suggesting that direct binding of EZH2 is involved in the regulation of p16^INK4a^. Therefore, overexpression of EZH2 seems to repress senescence-associated p16^INK4a^ transcription during cholangiocarcinogenesis. Similarly, EZH2 expression was associated with vascular infiltration, the histological grades, and the cell proliferation activity in mixed hepatocellular cholangiocarcinoma (HCC-CCA) [151].

Senescence can contribute to combined HCC-CCA development also through IL-6 signaling. In an inflammation-induced liver cancer mouse model, single-cell RNA sequencing analysis (scRNA-seq) revealed that HCC-CCA may originate from hepatic progenitor cells, whose transformation depends on IL-6 secreted by parenchymal and non-parenchymal liver senescent cells via the IL-6-gp130 pathway [152]. gp130 is a common signal-transducing component of the functional receptor complexes for the IL-6 family of cytokines, which can be expressed on the membrane of hepatic progenitor cells [153]. Senescence-induced IL-6 firstly binds to its soluble receptor, gp80, so that it can gain the affinity to be trans-presented to *gp130* receptor. Once IL-6 trans-signaling occurs, it induces the expression of pSTAT3 and pERK activity in pre-malignant cells, probably leading to tumor development [154]. Indeed, pSTAT3 and pERK were reported to be highly expressed in combined HCC-CCA tumors in both mice and humans [152]. Accordingly, the administration of anti-IL-6 antibodies hampered the development of combined HCC-CCA tumors. Thus, combined HCC-CCA originates from hepatic progenitor cells and, in the context of an inflammatory microenvironment, senescent-associated IL-6 plays an important role in tumor progression via IL-6 trans-signaling. Moreover, recent evidence showed that IL-6 secreted from cancer-associated fibroblasts (CAFs) triggers epigenetic changes in cholangiocytes, stimulating a malignant transformation towards iCCA [155]. Furthermore, the farnesoid X receptor (FXR), that is usually downregulated in iCCA cell lines and human samples, has been shown to act as a metastasis suppressor in this tumor by inhibiting IL-6-induced EMT [156]. Interestingly, IL-6 secreted by CAFs has been shown to induce chemotherapy resistance in CCA cells via IL-6/STAT3 activation, suggesting that senescence-derived IL-6 may exert a similar role [157]. However, contrasting evidence recently reported a negative correlation between IL-6 and iCCA [158]. Accordingly, it has been shown that the pharmacological inhibition of IL-6 trans-signaling via the administration of recombinant sgp130Fc increased tumor cell survival, proliferation, and migration, suggesting that IL-6 signaling can be associated with a better CCA prognosis [159]. In a carcinoma of the gallbladder (GBC), a CCA-related cancer, downregulation of the soluble receptor, gp80, positively correlated with an improvement in overall survival [153]. Taken together, these data suggest a crucial role of IL-6 in cholangiocarcinoma development, although it must be clarified whether it mainly exerts pro-tumorigenic or anti-tumorigenic functions.

The administration of therapies can also induce senescence in CCA cells. Chemotherapy, using gemcitabine and cisplatin (Gem+Cis), is the only approved treatment for advanced CCA, and is sometimes administered together with novel immunotherapy regimens [160,161]. Cisplatin, a platinum-based compound, acts by leading to severe DNA damage associated with DNA-crosslinking, which has been shown to result in senescence induction [162]. Meanwhile, gemcitabine can induce genotoxic stress by inhibiting DNA synthesis, thereby triggering cellular senescence [163]. The bile duct cancer drug, PRIMA-1^MET^, inhibits CCA cell growth via senescence induction. PRIMA-1^MET^ is a methylated derivative and structural analog of PRIMA-1, a cancer drug which triggers p53 re-activation [164]. The authors showed that two CCA cell lines (KKU-100 and KKU-21) treated with PRIMA-1^MET^ exhibited a senescent phenotype as visualized under a light microscope, including enlarged size, flattened cell shape, induced nuclear granularity, and increased volume of cytoplasm, together with β-gal expression. Moreover, treated cells showed a significant increase in p21 and p16^INK4A^ expression compared to untreated cells. Interestingly, these results were confirmed by immunohistochemical analysis of CCA tissues from 160 patients with intrahepatic CCA. Being p21 and p16^INK4A^ markers associated with good prognosis in different cancers, such as adenocarcinoma and Hodgkin lymphoma, these results suggest that patients with CCA who have a high expression of these markers might be predicted as belonging to a good prognostic group [165]. Further evidence showed that CCA cell lines treated with five different compounds (5-aza-2′deoxycytidine, bromodeoxyuridine, interferons IFNβ and IFNγ, as well as hydrogen peroxide) underwent senescence [166]. Indeed, treatment with all five agents decreased cell proliferation and induced cellular senescence, with different degrees of growth-inhibitory effects depending on the cell type and origin, and displayed overexpression of p21 and interferon-related genes. In addition, the genotoxic drug, bromodeoxyuridine, displayed a stronger induction of senescence, suggesting that CCA cells’ senescent state is mainly controlled by DNA damage response pathways relating to p53/p21 signaling. Finally, IFNβ and IFNγ treatment triggered senescence, thus suggesting a possible involvement of IFN pathways in the achievement of this cellular condition. Overall, these results indicate that theinduction of cellular senescence can be a promising therapeutic strategy for inhibiting the growth of CCA cells, allowing the development of novel therapeutic strategies for patients’ treatment.

### 4.3. Senescence in Colorectal Liver Metastases (CLM)

Colorectal cancer (CRC) is the third most commonly diagnosed malignancy and the second leading cause of cancer-related deaths in the world [167]. Despite significant improvements in diagnosis and treatment, most deaths are caused by the development of distant metastasis, highly resistant to therapies [168,169]. Indeed, about half of patients affected by CRC already have colorectal liver metastasis (CLM) at the time of diagnosis, or they will develop them after the resection of the primary tumor with an extremely poor 5-year survival rate [170]. The liver is the most common CRC metastatic site because it receives and filters the blood from the intestine through the portal vein [171]. CLM development is a multistep biological process, where CRC cells’ dissemination from the primary site to the liver is possible due to various mechanisms, including EMT and angiogenesis [172]. Firstly, cancer cells migrate to the tissues around the primary CRC site and then they spread in venules, capillaries, and lymphatic vessels until they enter the systemic circulation that will pave their way towards the liver [173]. The EMT mechanism allows tumor cells to acquire a wide range of biological and molecular changes, which facilitate their dissemination from the primary site to secondary metastatic site in distant organs [174]. CLMs’ spread is also enhanced by the angiogenesis process, although CRC cancer cells can also metastasize by hijacking pre-existing vessels of the host liver when the conditions are not favorable to form new ones [175]. This non-angiogenic process is referred to as vessels’ co-option [176]. The metastatic process is fostered by an intricate crosstalk between CLM cells and the highly heterogeneous and suppressive immune microenvironment [177]. A wealth of studies showed that the heterogeneous population of tumor-associated macrophages (TAMs) may provide a nurturing environment for the development of CLMs and play a pivotal role in the efficacy of anticancer strategies [178]. Intercellular networking between macrophages and fibroblasts has been shown to support CRC, enhancing the immunosuppressed metastatic niche in the liver by leading to a dysfunctional state of CD8+ T lymphocytes [179]. Accordingly, high densities of tumor-infiltrating lymphocytes (TILs) in CRC primary tumors are associated with a good prognosis and responsiveness to chemotherapy [180]. Moreover, scRNA-seq uncovered malignancy-associated exhausted and regulatory T cells, together with a subset of dendritic cells (DC3), as players involved in CLM development [181]. Therefore, the host immune system represents one of the key factors in determining the patient’s prognosis, and its involvement in the study of new effective immunotherapies is rapidly evolving [182]. Nowadays, curative tumor resection and chemotherapy are the standard methods of treatment in patients with CLMs [183]. However, surgery is only possible in 10–20% of cases, depending on the site and size of the tumor, the presence of extrahepatic diseases or patient comorbidities [184]. Despite the significant progress in the development of new chemotherapeutic drugs for CLMs, patients who receive fluorouracil and platinum chemotherapy may develop resistance [185]. Furthermore, even in patients treated with immune checkpoint inhibitors, such as anti-PD-1/PD-L1 drugs, the presence of liver metastases was associated with resistance compared to primary CRC [186]. The main causes of therapy resistance in metastatic CRC have been associated with numerous single-nucleotide polymorphisms (SNPs), which help to explain the genetics behind these mechanisms of resistance and the heterogeneity of therapy response among patients. SNPs, indeed, may confer resistance to chemotherapy and ICB inhibitors by inducing decreased intracellular drug concentration, altered metabolism, or alterations to the targets of the therapy [187]. However, resistance is more complex and does not involve mutation but molecular pathway alteration, including crosstalk between associated pathways which activate complementary cell survival and growth mechanisms [188]. For these reasons, further elucidation of the mechanisms underlying therapy resistance in CLMs is required. Discoveries will continue to translate into improved treatment options and important clinical outcomes for patients.

Senescent cancer cells accumulate in CRC, favoring tumor dissemination. Although studies on CRC metastatic patients are still rare, senescence has been identified to play a crucial role in CLMs. Haugstetter et al. described cellular senescence as a positive prognostic factor in CLM patients’ treatment and survival [189]. Contrarily, senescent cancer cells have been found to support tumor spread in CRC [190]. Exploiting both human and mouse samples, Braumüller et al. showed that SASP factors may lead to enhanced immuno-senescence in the TME of CRC, thus explaining its resistance to known chemo- and immunotherapeutic approaches [191]. Indeed, the authors showed that senescent cancer cells acquired a stem cell-like phenotype in both murine and human samples, possibly leading to deleterious consequences, like chemotherapy resistance. SASP factors released by senescent tumor cells in the murine model induced infiltrating T-cells to upregulate checkpoint inhibitor molecules as well as senescence-associated molecules, indicating that T-cells were not exhausted but senescent. In human counterparts, tumor-infiltrating immune cells showed upregulation of SA-β-gal activity, while non-immune cells showed no senescence induction. This phenomenon of immune senescence can lead to therapeutic difficulties as senescent immune cells are unaffected by checkpoint inhibitor therapy. Moreover, various SASP factors, including interleukins 1, 6 and 8, as well as chemokines CXCL8 and CCL2 have been associated with accelerated tumor proliferation and invasion in CRC [95]. Furthermore, age-related senescence of human peritoneal mesothelial cells has been shown to create a more permissive environment for the metastatic colonization of the peritoneum in CRC, enhancing circulating tumor cells’ adhesion, especially via ICAM-1 (intercellular adhesion molecule 1) [192]. Accordingly, co-injection of CRC cells together with senescent peritoneal mesothelial cells led to the upregulation of EMT, angiogenesis, and tumor progression mediated by an increased secretion of proinflammatory molecules from peritoneal mesothelial senescent cells, which can perform a crucial role in the metastatic process [193]. Taken together, these findings underline the conflicting role of cellular senescence in cancer dissemination from the primary organ to the metastatic site. Interestingly, these contrasting results have been recently clarified by Garbarino et al., who discovered two distinct senescent metastatic cancer cell (SMCC) subtypes in CLMs, transcriptionally located at the opposite pole of EMT [194]. Epithelial and mesenchymal subtypes are characterized by different biological behaviors and secretome, responses to chemotherapy, and prognoses. The epithelial senescent CLM state has been shown to be triggered by nucleolar stress. Indeed, the hyperactivation of c-Myc, an oncogene involved in ribosome biogenesis, can induce the accumulation of ribosomal protein, RPL11, and the DDR. Spatial transcriptomics analysis performed on CLM sections belonging to post-chemotherapy patients, showed that the accumulation of RPL11 may inhibit the interaction between p53 and its E3 ubiquitin-protein ligase, HDM2, leading to senescence activation in epithelial senescent CLM cells. Similar results have been achieved by inducing senescence in HCT-116 cells treated with Doxorubicin, a chemotherapy agent also known to induce potent ribosomal stress. After its administration, the authors highlighted the activation of a p53-dependent senescence phenotype in HCT-116-treated cells and showed that the transcriptional signature of these cells overlapped with both epithelial and mesenchymal senescent CLM cells, supporting the hypothesis that the type of stressor rather than the cell type of origin is determinant of the senescent phenotype. Contrarily, the TGFβ paracrine activation of NADPH oxidase 4 (NOX4)-p15 effectors has been shown to drive the mesenchymal senescent CLM signature. Mechanistically, TGFβ may inhibit proliferation by inducing G1-phasecell-cycle arrest through the activation of NOX4 and accumulation of reactive oxygen species. These conditions lead to c-MYC downregulation, which in turn enhance the expression of CDK inhibitory genes, p15^Ink4b^ and p21^Cip1^. Furthermore, NOX4 activation as a downstream effector of TGFβ1 is associated with other known metastasis-promoting pathways, including EMT, angiogenesis, and tissue remodeling. Accordingly, patients displaying a higher expression of epithelial senescence CLM markers displayed better prognoses rather than mesenchymal ones, suggesting a crucial role of senescence in CLM development, prognosis, and therapy response.

As it happens for HCC and CCA, anticancer therapies can enhance the senescence process in CLM. Chemotherapy has been shown to accelerate immune-senescence and functional impairments of Vδ2^pos^ T cells, a specific human γδ T lymphocyte subset involved in cancer immune-surveillance with promising perspectives in cancer immunotherapy [195]. Specifically, the analysis of the peripheral blood of CLM patients who had undergone chemotherapy showed a decreased amount of Vδ2^pos^ T cells, while terminally differentiated CD27neg/CD45RApos effector memory T cells (TEMRA) increased. This latter population expresses senescent marker, CD57, together with functional impairment in cytotoxicity and the production of TNF-α and IFN-γ, thus suggesting the acquisition of an immune-senescent profile in CLM patients after chemotherapy treatment. Similar results were achieved via the analysis of tumor-infiltrating Vδ2^pos^ T cells purified from CLM specimens of patients who had undergone chemotherapy. Overall, this evidence shows that chemotherapy treatment can induce the accumulation of highly dysfunctional Vδ2^pos^ T cells, impairing their antitumor activity. Moreover, other recent evidence showed that senescent rectal cancer cells can secrete SASP which may trigger EMT both in vitro and in clinical samples from patients with rectal cancer who had undergone neoadjuvant chemoradiotherapy [196]. Indeed, cultured colon cancer cells (HCT 116) induced into senescence via exposure to 5-fluorouracil 5-FU can lead to EMT in colon and rectal cancer cell lines and increase cell invasion in vitro. Similar observations were obtained by analyzing rectal cancer samples from patients treated with neoadjuvant chemotherapy, in which tumor cell niches were enriched for senescent cells characterized by increased mRNA expression levels of EMT-related proteins (Slug, Snail, vimentin) compared with non-senescent niches. Finally, these results suggest that senescent cancer cells can influence the TME by promoting immune evasion and EMT thereby enhancing metastatic spread in CLMs.

## 5. Conclusions and Future Challenges: How Senescence Can Be Exploited for Liver Cancer Treatments

Cancer therapies causing high levels of DNA damage are still the primary treatment options for many tumors, but they can lead to senescence induction and SASP factors’ secretion, whose persistence after treatment may be detrimental and enhance tumor growth [17]. Therefore, a strategy in which therapy-induced senescent cells are eliminated is needed for minimizing the risk of tumor progression and avoiding adverse effects. Senotherapy represents a new therapeutic approach, which involves the selective elimination of senescent cells using drugs called senolytics or to the limitation of SASP factors’ production and secretion by exploiting agents termed senomorphics [16]. The combination of senescence-inducing chemotherapies followed by senolytics has been shown to increase tumor cell killing and/or elimination of the remaining disease [197,198]. Indeed, senolytic drugs mainly target antiapoptotic pathways that senescent tumor cells exploit to survive despite the increased stress signaling [18]. Among promising senolytics drugs for liver cancer treatment, recently, the bromodomain and extra-terminal domain family protein degrader (BETd) has been identified, whose administration has been shown to induce the elimination of senescent hepatic stellate cells in obese mouse livers, which led to a reduction in liver cancer development and an increase in chemotherapy response [199]. The Bcl2 family antagonist, ABT-263 (Navitoclax), has been shown to limit tumor development by selectively inducing the apoptosis of senescent hepatic stellate cells, known to contribute to HCC growth [200]. The administration of the combined senolytic drugs, dasatinib and quercetin (D+Q), reduced inflammation and HCC in the liver of an aging mouse model [201]. However, a D+Q senolytic cocktail has been shown to be ineffective in enhancing the efficacy of senescence-inducing chemotherapy and appeared to have acute pro-tumorigenic effects in non-alcoholic fatty acid liver disease (NAFLD) mice [202]. Besides senolytics, also senomorphics have shown promising anticancer results in liver cancers. The mTOR pathway is an essential SASP regulator, and mTOR inhibitors have shown good effects in reducing senescent liver cancer cells. In a mouse model of liver cancer, the mTOR inhibitor, AZD8055, showed effective senolytic potential against senescent liver cancer cells [198]. Additionally, CFI-402257, an inhibitor which targets cell-cycle checkpoints, induced the release of specific SASPs that further attracted various subsets of immune cells (NK cells, CD4+ T cells, and CD8+ T cells) to control HCC [203]. Even in combined HCC-CCA tumors, treatment with senolytic agents resulted in the inhibition of both IL-6 SASP and related tumor growth [152]. These results reveal the vulnerability of senescent cells and show possibilities for their control.

Despite promising results in HCC clinical trials, there are toxicities associated with these compounds, thereby limiting their clinical applications [203]. Furthermore, the lack of specific senescence markers and the difficulty in understanding the dynamic role of senescence in different disease stages, contexts, and cell types challenges the use of senescence as a weapon to target HCC. Finally, the role of these drugs in other liver cancers, including CCA and CLM, as well as in precancerous and early stages of HCC, still needs to be investigated. The elucidation of the role of cellular senescence as a therapeutic target for cancer treatment will enable the translation of senolytics into the clinic in the near future, ameliorating patients’ management and providing more effective treatments against the different types of liver cancer.

## Figures and Tables

**Figure 1 biomedicines-12-00026-f001:**
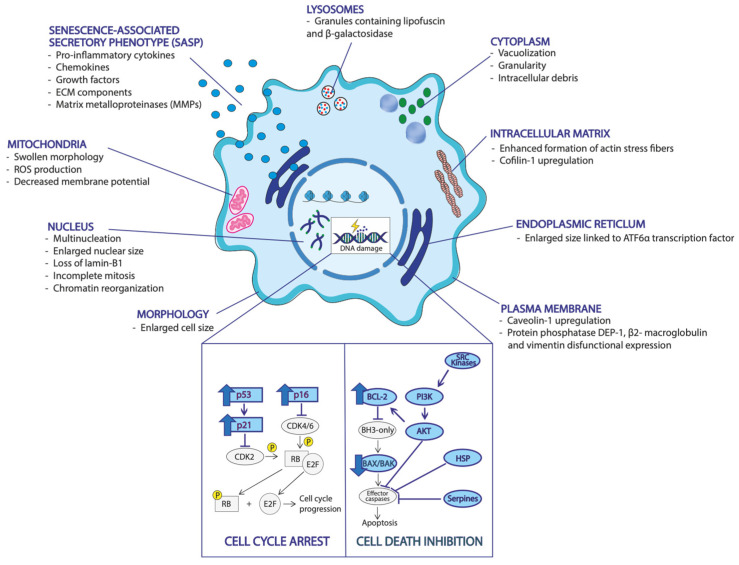
General hallmarks of senescence. The senescent state is characterized by a wide variety of morphological features, which include aberrations and changes in cellular morphology, plasma membrane, nuclear composition and cytoplasm granularity and vacuolization. Senescence mechanism affects the conditions of different organelles, including mitochondria, endoplasmatic reticulum, and lysosomes. Lysosomal granules contain lipofuscin and β-galactosidase (SA-β-GaSenescence), which represent the most important senescent marker. The senescent state is often triggered by DNA damage, starting multiple pathways responsible for the inhibition of cell cycle and the escape from cell death, thus conferring to senescent cells the ability to survive even if it does not replicate. This feature explains the reason why senescent cells are often referred to as “zombies”.

**Figure 2 biomedicines-12-00026-f002:**
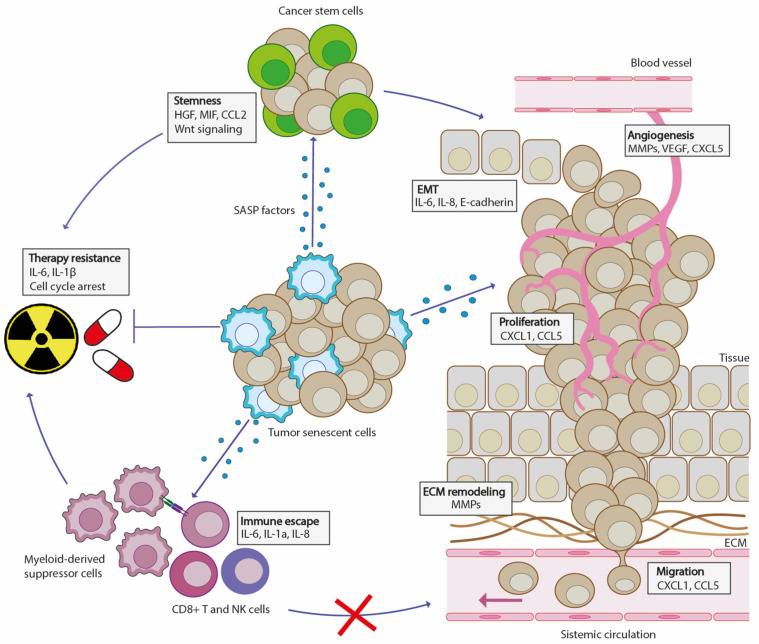
The role of senescence in tumor development. When it occurs in cancer, senescence provides a wide variety of SASP factors involved in tumor promotion. The release by senescent tumor cells of HGF, MIF and CCL2 factors enforces tumor cells’ stemness, generating aggressive tumor clones which may become therapy-resistant. Besides stemness, senescence is also involved in immune escape, another feature that fosters resistance to anticancer treatments. Indeed, IL-6, IL-1a, and IL-8 SASP factors promote the recruitment of myeloid-derived suppressor cells, which support tumor-growth by inhibiting CD8+ T cells and NK cells’ antitumor activities. Moreover, SASP factors are involved in various tumor-promoting mechanisms, including epithelial-to-mesenchymal transition (EMT), angiogenesis, proliferation, ECM remodeling, and migration. EMT: epithelia-to-mesenchymal transition; ECM: extracellular matrix.

**Figure 3 biomedicines-12-00026-f003:**
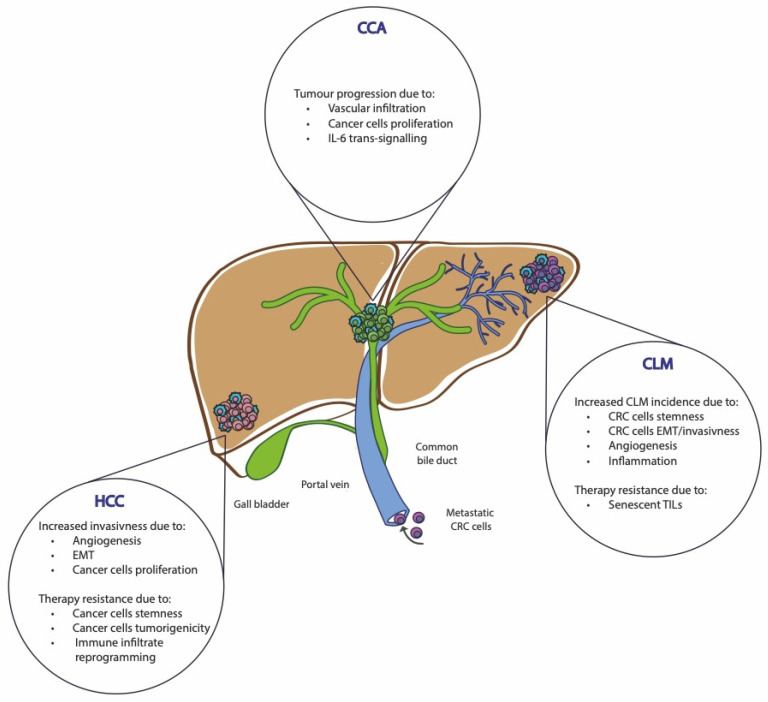
Pro-tumor functions of senescence in liver cancer. Senescence mechanism has been associated with tumor progression in most liver tumors. In cholangiocarcinoma, senescent cells have been observed to induce vascular infiltration and cancer cell proliferation. In hepatocellular carcinoma, the senescence phenotype induces an increased invasiveness due to angiogenesis, EMT, and cancer cell proliferation. Increased colorectal liver metastases incidence has been associated with senescence, which has been associated with stemness, EMT, angiogenesis, and inflammation. The ability of senescence cells to foster tumor growth and invasiveness has been observed to induce resistance to therapies, mainly due to the acquisition of stemness and immune escape. CCA: cholangiocarcinoma; HCC: hepatocellular carcinoma; CLM: colorectal liver metastases; EMT: epithelial-to-mesenchymal transition; TILs: tumor-infiltrating lymphocytes.

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
