# Peer review of "Cellular Senescence in Liver Cancer: How Dying Cells Become “Zombie” Enemies"

_biomedicines, 2023, doi:10.3390/biomedicines12010026_

Round 1
Reviewer 1 Report
Comments and Suggestions for Authors
Dear editors:
It is a great honor and pleasure for me to be invited as the reviewer for this important work entitled “Cellular Senescence in Liver Cancer: How Dying Cells Become 2 “Zombie” Enemies”. Aurora Gazzillo and co-authors comprehensively reviewed the recent evidence of senescence-associated hepatocarcinogenesis and associated therapies. This study topic is novel and advanced, attributing to corresponding author Matteo Donadon’s long-term efforts and contributions in this scientific field. I have a few comments concerning this study:
The conclusion section should be more concise.
“Recent evidence using an animal model that fully recapitulates non-alcoholic fatty acid liver disease (NAFLD) 772 showed that D + Q senolyitic cocktail is ineffective against age-associated NAFLD-induced HCC [204]. Similarly, studies performed on HCC cell lines and xenograft HCC mouse models demonstrated that D + Q not only wasn’t able to enhance the efficacy of senescence-inducing chemotherapy, but also D + Q alone appeared to have acute pro-tumorigenic effects in control mice [205].” The above cocktail therapy is not mentioned in the text that should be described in other section. The references might not exist in the conclusion section.
The research is interesting that should be published after appropriate revision.
Comments on the Quality of English LanguageMinor editing of English language is required.
Author Response
Reviewer #1: Dear editors,
It is a great honor and pleasure for me to be invited as the reviewer for this important work entitled “Cellular Senescence in Liver Cancer: How Dying Cells Become “Zombie” Enemies”. Aurora Gazzillo and co-authors comprehensively reviewed the recent evidence of senescence-associated hepatocarcinogenesis and associated therapies. This study topic is novel and advanced, attributing to corresponding author Matteo Donadon’s long-term efforts and contributions in this scientific field.
Reply: We thank the reviewer for the positive comments.
I have a few comments concerning this study: The conclusion section should be more concise.
Reply: We thank the reviewer for this suggestion. The text of paragraph 5 has been revised to make the conclusion section more concise. The changes have been implemented using the Word “Track Changes” function.
“Recent evidence using an animal model that fully recapitulates non-alcoholic fatty acid liver disease (NAFLD) 772 showed that D + Q senolyitic cocktail is ineffective against age-associated NAFLD-induced HCC [204]. Similarly, studies performed on HCC cell lines and xenograft HCC mouse models demonstrated that D + Q not only wasn’t able to enhance the efficacy of senescence-inducing chemotherapy, but also D + Q alone appeared to have acute pro-tumorigenic effects in control mice [205].” The above cocktail therapy is not mentioned in the text that should be described in other section. The references might not exist in the conclusion section. The research is interesting that should be published after appropriate revision.
Reply: We thank the reviewer for this comment. The text of paragraph 5 has been revised to provide a short section devoted to the D+Q cocktail therapy and its related toxicity in NAFLD prior to the conclusion section. Moreover, the references have been moved from the conclusion to previous paragraphs as suggested. The changes have been implemented using the Word “Track Changes” function.
Minor comments
Comments on the Quality of English Language: Minor editing of English language is required.
Reply: We thank the reviewer for the useful suggestion. Grammar and spelling errors were entirely revised as suggested. The change has been implemented using the Word “Track Changes” function.
Reviewer 2 Report
Comments and Suggestions for Authors
In the present review, the authors intend to provide an overview of the recent evidence that unveils the role of cellular senescence in the most frequent forms of primary. As presented in the manuscript, liver cancer represents the fourth leading cause of cancer-associated death worldwide. The heterogeneity of its tumor microenvironment (TME) is a major contributing factor to metastasis, relapses, and drug resistance. Late diagnosis makes most liver cancer patients ineligible for surgery and the frequent failure of non-surgical therapeutic options orientates clinical research to the investigation of new drugs. In this context, cellular senescence has been recently shown to play a pivotal role in the progression of chronic inflammatory liver diseases, ultimately leading to cancer. Moreover, the stem-like state triggered by senescence has been associated with the emergence of drug-resistant, aggressive tumor clones. In recent years, an increasing number of studies have emerged to investigate senescence-associated hepatocarcinogenesis and its derived therapies, leading to promising results. Based on the presented manuscript, I suppose that this work should be accepted for publication in the present form - however, I propose a criterion revision of the text because minor mistakes were found in some sentences.
Comments on the Quality of English LanguageA revision in the text is necessary because minor mistakes were found in some sentences.
Author Response
Reviewer #2: In the present review, the authors intend to provide an overview of the recent evidence that unveils the role of cellular senescence in the most frequent forms of primary. As presented in the manuscript, liver cancer represents the fourth leading cause of cancer-associated death worldwide. The heterogeneity of its tumor microenvironment (TME) is a major contributing factor to metastasis, relapses, and drug resistance. Late diagnosis makes most liver cancer patients ineligible for surgery and the frequent failure of non-surgical therapeutic options orientates clinical research to the investigation of new drugs. In this context, cellular senescence has been recently shown to play a pivotal role in the progression of chronic inflammatory liver diseases, ultimately leading to cancer. Moreover, the stem-like state triggered by senescence has been associated with the emergence of drug-resistant, aggressive tumor clones. In recent years, an increasing number of studies have emerged to investigate senescence-associated hepatocarcinogenesis and its derived therapies, leading to promising results. Based on the presented manuscript, I suppose that this work should be accepted for publication in the present form - however, I propose a criterion revision of the text because minor mistakes were found in some sentences.
Minor comments
Comments on the Quality of English Language: a revision in the text is necessary because minor mistakes were found in some sentences.
Reply: We thank the reviewer for the positive comments. The text of the manuscript has been entirely revised to check all the grammar and spelling error present in some sentences. The change has been implemented using the Word “Track Changes” function.